# Retrospective Analysis of Ultrasound-Guided Serial-Injection Triple Nerve Block Efficacy in Cementless Bipolar Hemiarthroplasty for Femoral Neck Fracture

**DOI:** 10.3390/jcm13020338

**Published:** 2024-01-07

**Authors:** Jung Wook Huh, Min Woo Kim, Young Min Noh, Han Eol Seo, Dong Ha Lee

**Affiliations:** Department of Orthopedic Surgery, Busan Medical Center, Busan 47527, Republic of Korea; gizer00@hanmail.net (J.W.H.); drkimminwoo@naver.com (M.W.K.); doctornoh77@naver.com (Y.M.N.); mdseo86@gmail.com (H.E.S.)

**Keywords:** femur neck fracture, bipolar hemiarthroplasty, ultrasonography, peripheral, nerve block

## Abstract

Background: Femoral neck fractures are effectively treated with bipolar hemiarthroplasty (BHA) surgery, yet postoperative pain management remains a challenge. This study explores the efficacy of multimodal pain management in minimizing opioid use and enhancing recovery. Methods: A retrospective analysis of 87 patients who underwent BHA between September 2016 and September 2020 was conducted. Patients were analyzed in two groups: Group I (*n* = 42), receiving serial-injection nerve blocks (SINBs) before and after surgery, and Group II (*n* = 41), with no SINB. Notably, all nerve blocks for Group I were performed after November 2017, following the implementation of this technique in our protocol. Pain and analgesic medication usage were assessed over 72 h post-surgery, along with hospitalization duration and perioperative complications. Results: Group I patients exhibited significantly lower pain scores at 6, 12, 24, and 48 h post-surgery, alongside reduced incidences of postoperative nausea and vomiting (PONV) and delirium compared with Group II (*p* < 0.05). Conclusions: Utilizing sequential lower limb nerve blocks under ultrasound guidance in BHA surgeries effectively reduces early postoperative pain and associated adverse effects. This approach demonstrates potential benefits in pain management, leading to diminished narcotic usage and lower risks of PONV and delirium.

## 1. Introduction

Bipolar hemiarthroplasty (BHA) is a common surgical intervention used to manage the acute pain associated with femoral neck fractures. While effective, BHA is linked to significant postoperative pain that, if not well-managed, can hinder recovery, lengthen hospitalization, and increase the risk of negative outcomes such as myocardial ischemia, pulmonary dysfunction, and thromboembolism. Therefore, optimal pain management after BHA should prioritize providing potent pain relief while minimizing opioid consumption and supporting recovery. Similarly, total knee arthroplasty (TKA) can also result in significant postoperative pain, and various pain management strategies are available, such as systemic or intrathecal opioids, local infiltration analgesia, and peripheral nerve blocks (PNBs). However, the ideal pain management approach should offer robust analgesia without any unwanted consequences to minimize adverse effects [1,2,3]. 

The hip joint receives innervation from multiple nerves. Specifically, the obturator nerve and articular branches of the femoral nerve innervate the anteromedial section of the hip joint, while the lateral femoral cutaneous nerve (LFCN) innervates the anterolateral section [4]. It has been observed that patients undergoing a direct lateral approach for femur neck fracture at our institution often report postoperative pain in both the anteromedial and anterolateral regions of the hip joint [5], which can be attributed to the involvement of these nerves. 

Inserting a catheter that can cover all three nerves is a challenging procedure, mainly due to the complexity of the process and the location of the nerves. The femoral nerve seems to be the most feasible option, but the success of the blocks using this technique is unpredictable [4]. Using ultrasound guidance can make the procedure easier, but the results may still be unreliable. The procedure is technically difficult, even for experienced practitioners, as it requires penetrating a needle through two fascial layers [6].

The fascia iliaca compartment block (FICB) is a well-established technique for pain management in hip and thigh surgeries [7,8]. It involves injecting local anesthetics into the fascia iliaca compartment to block the femoral, lateral femoral cutaneous, and obturator nerves. Comparatively, the pericapsular nerve group (PENG) block, a newer approach, targets sensory nerves around the hip joint, particularly the articular branches of the femoral nerve and the accessory obturator nerve, and is gaining recognition for its effectiveness in hip surgery analgesia [9].

In the context of total hip arthroplasty, Capdevila et al. recommend using a continuous psoas compartment block for total hip arthroplasty, utilizing a modified Winnie’s landmark [10] to accurately determine the distance between the lumbar plexus and L4 transverse process [11]. They found that the fascia iliaca compartment block is more effective than the three-in-one block. However, it is worth noting that both techniques provide sensory blockade in only 35% of cases [12]. Therefore, we opted to perform a single injection of each of the three nerves before the BHA procedure, rather than dwelling a catheter on one nerve that innervates the hip joint.

Ensuring that the effects of peripheral nerve blocks are sustained is crucial to reducing opioid use among inpatients, as even small doses (20 to 50 MME/day) may increase the risk of clinical complications and long-term opioid dependence [7,13,14]. However, the duration of a single-injection nerve block is limited to a maximum of 12 h [15]. To extend the effects beyond 12 h, we developed a method of serial injection.

The purpose of this investigation was to evaluate the clinical benefits of ultrasound-guided serial-injection nerve blocks (SINBs) targeting the femoral, obturator, and lateral femoral cutaneous nerves in patients undergoing BHA. As these peripheral nerves innervate the proximal femur and hip joint, this study aimed to assess the effectiveness of SINBs in achieving optimal pain relief while minimizing opioid consumption during the first 48 h postoperatively. Furthermore, we aimed to investigate the extent to which SINBs decrease the incidence of postoperative nausea and vomiting (PONV) and delirium, thereby promoting enhanced rehabilitation.

## 2. Materials and Methods

### 2.1. Trial Design and Study Settings 

A direct lateral approach under spinal anesthesia was utilized by a single orthopedic surgeon to perform all BHA procedures in our study. Our study population consisted of 83 patients who underwent BHA between September 2016 and September 2020, all of whom were operated on by the same surgeon. These patients were divided into two groups—those who received SINB and those who did not. The SINB procedure was performed for all patients after January 2018, spanning a period of 2 years and 7 months.

During the BHA procedure, all patients received a standard dose of 0.2 mL mg/kg of bupivacaine (Heavy Marcaine^TM^, AstraZeneca, London, UK) intrathecally under spinal anesthesia.

### 2.2. Participants

In our study, we conducted a retrospective review of the medical records of 87 patients who had undergone BHA for fragility femoral neck fracture at Busan Medical Center, Busan, South Korea, between September 2016 and September 2020. This study was approved by the Public Institutional Bioethics Committee designated by the MOHW, and the Institutional Review Board granted approval under the number P01-202011-21-027.

To ensure the validity of our results, patients were excluded from this study if they met certain criteria, including pre-existing immobility prior to the injury, malunion or nonunion resulting from delayed initial treatment, diagnosis of severe dementia or other psychiatric conditions, or signs of delirium within 24 h of surgery (a delirium rating scale of >10). In total, four patients were excluded from this study, as shown in Figure 1.

### 2.3. Interventions 

An experienced orthopedic surgeon performed an ultrasound-guided block of the lateral femoral cutaneous nerve, obturator nerve, and femoral nerve. The procedure took less than ten minutes for each nerve block, and it did not significantly add to the perioperative time. 

Group 1 comprised the intervention group receiving the saphenous, femoral, and lateral femoral cutaneous nerve blocks (SINBs), while Group II was designated as the control group and did not receive any blocks. All SINBs were performed by a single experienced orthopedic surgeon immediately prior to surgery and repeated thrice postoperatively at 12 h intervals. A 5 cm long, 5–12 MHz linear probe (LOGIQ e, GE, Boston, MA, USA) and 22G spinal needle were used for the nerve blocks under ultrasound guidance. The femoral, obturator, and lateral femoral cutaneous nerves were each blocked with 7.5, 4, and 4 mL of 0.75% ropivacaine mixed with 7.5, 4, and 4 mL of 1% lidocaine in a 1:1 ratio, respectively, which is similar to the method used in a previous study. A sensory test including a pinprick test was performed to confirm the efficacy of the blocks. Premedication included 200 mg of Celebrex (celecoxib; Pfizer, New York, NY, USA), and no additional local anesthetics or analgesics were given during the surgery.

Postoperative pain management included patient-controlled intravenous analgesia, consisting of a mixture of 2 mg of butorphanol tartrate (Myungmoon Pharm., Seoul, Republic of Korea), 50 mg of tramadol hydrochloride (Yuhan Corp., Seoul, Republic of Korea), and 30 mg of ketorolac tromethamine (Hanmi Pharm., Seoul, Republic of Korea) mixed in 100 mL of saline, with a background infusion rate of 0.05 mL/hr, bolus of 0.2 mL, and lockout of 8 min. This regimen was calculated as 19 MME (morphine milligram equivalents), and the number of doses administered was recorded by checking the volume remaining on the 3rd postoperative day. Rescue analgesics (50 mg of tramadol or 90 mg of diclofenac) were administered to patients as required for postoperative pain control, and anti-emetic injection was only given to those who exhibited symptoms of nausea and vomiting.

### 2.4. Outcome Assessments

The postoperative pain intensity was assessed using the visual analog scale (VAS) at 6, 12, 24, 48, and 72 h after the surgery. The medical staff provided a detailed explanation of the meaning of the VAS before the initial measurement. The number of doses administered via patient-controlled analgesia (PCA) was recorded for the first 72 h, along with the amount of rescue analgesics. Additionally, the number of cases of nausea and vomiting was recorded from the electronic medical records. The Koval classification grade (ambulatory ability I-VII), T-cane walking (days until the patient was able to walk with the aid of a T-cane after the operation), local complications due to SINB, general postoperative complications (pressure sore, pneumonia, deep vein thrombosis, postoperative nausea and vomiting, and delirium), and duration of hospitalization were also compared.

### 2.5. Statistical Methods 

The statistical analysis was carried out using the SPSS version 22.0 software (IBM Corporation, Armonk, NY, USA). Patient characteristics were evaluated using frequency analysis and cross-analysis. The t-test was used to determine any differences in clinical outcomes and patient demographics between the two groups. Power analysis was conducted, which revealed an effect size of 0.5, a statistical significance level of 0.05, and a statistical power of 0.90 for both groups.

## 3. Results

The two groups did not differ significantly in terms of age (Group I: 57–88; Group II: 59–86), gender, sex, BMI (kg/m^2^), Charlson Comorbidity Index (CCI), VAS measured in the ward on the day of admission, Koval classification grade, blood loss, or urine output. However, the operative time was longer in Group II by an average of 8.58 min (Table 1). Group I had significantly lower subjective pain scores compared with Group II at 6, 12, 24, and 48 h after BHA (*p* < 0.05). However, there were no significant differences in postoperative VAS scores at 72 h (Table 2).

In terms of pain management, Group I used a smaller volume of the PCA solution (65.72 mL; morphine equivalent dose: 82.8) in the first 72 h after surgery compared with Group II (83.90 mL; morphine equivalent dose: 95.8). The number of injections administered via PCA was also significantly lower in Group I, with an average of 0.65 injections for Group I patients and 0.88 injections for Group II patients (Table 3).

Furthermore, there were no local complications, such as infection or hematoma, at the injection site. In addition to the potential temporary weakness in the quadriceps muscle, a known side effect of femoral nerve blocks [13,14,15], we made a concerted effort to preserve ankle dorsiflexion and plantarflexion to mitigate the risk of deep vein thrombosis (DVT) during our study. Patients were instructed on straight leg raising exercises, knee flexion/extension, and ankle dorsiflexion/plantarflexion exercises prior to surgery and encouraged to continue these exercises postoperatively to facilitate rehabilitation. Additionally, patients were educated on postural changes and were provided with medical compression stockings throughout their hospital stay to help prevent postoperative complications.

The incidence of pressure sores, pneumonia, and deep vein thrombosis did not differ significantly between the groups. However, the incidence of postoperative nausea and vomiting (PONV) (0.00%) and delirium (0.07%) in Group I was lower than that in Group II (0.15% and 0.24%, respectively). There were no reports of any local complications due to SINBs or any reports of block failure. The lengths of hospital stays and T-cane walking start days after the operations did not differ significantly between the groups (Table 4).

## 4. Discussion

The limitation of a single-injection nerve block is that it can manage postoperative pain after BHA only up to 12 h. Serial-injection nerve blocks effectively extended the duration of analgesic effects necessary for postoperative pain management. In this study, SINBs achieved adequate pain control not only for 12 h after surgery but also up to 48 h after surgery. Besides pain control, general complications such as the incidence of PONV and delirium were remarkably reduced due to the SINBs’ effect. These results are quite different from other studies, which demonstrated no significant difference in the delirium rate with CFNC (continuous femoral nerve catheter) treatment [14]. 

In addressing the potential influence of pain management on the incidence of delirium, it is imperative to first acknowledge the complexity of delirium’s etiology. Delirium is a multifaceted syndrome, influenced by numerous factors including, but not limited to, the patient’s age, underlying medical conditions, environmental factors, and pharmacologic agents [16,17]. Among these, the role of pain and pain management strategies, especially in postoperative settings, is increasingly recognized as a significant contributor.

The traditional understanding, supported by various studies, posits that uncontrolled pain is a critical risk factor for the development of delirium. Pain, particularly in the postoperative period, can be a source of significant physiological and psychological stress, potentially triggering or exacerbating delirious states. Conversely, effective pain control is often associated with reduced rates of delirium, as it alleviates one of the key stressors in the postoperative environment [18]. However, the relationship between pain management, particularly opioid use, and delirium is paradoxical. While opioids like morphine are effective for pain relief, they also carry a risk of inducing or worsening delirium, particularly in vulnerable populations such as the elderly.

Our results suggest that the method of pain management may play a crucial role in influencing delirium rates. The SINB technique’s ability to provide effective analgesia while potentially mitigating the side effects associated with systemic opioids could be a key factor in reducing delirium incidence. This aligns with the growing body of evidence suggesting that multimodal pain management strategies, which minimize opioid use, could be beneficial in managing postoperative delirium. It is important to note, however, that while our findings are promising, they also highlight the need for further research to fully understand the dynamics between pain management techniques and delirium incidence.

Given the complexity of delirium and the myriad factors contributing to its development, it is indeed remarkable that a singular intervention like effective pain control can have such a profound impact [17,19]. Our study contributes to the evolving understanding of postoperative delirium and underscores the importance of pain management not only for its primary purpose but also for its potential role in mitigating complex postoperative complications like delirium.

Performing triple nerve blocks every 12 h for two days can be a labor-intensive process. To address this issue, we implemented a manual approach for serial triple nerve blocks. Prior to the BHA surgery, triple nerve blocks were performed with ultrasound guidance just before spinal anesthesia in the operating room. Following the surgery, we conducted rounds approximately 12 h later (following the first triple nerve blocks) and performed the triple nerve block at the patient’s bedside with the assistance of the ward nurse. On postoperative day 1, we conducted wound dressing and repeated the triple nerve blocks at the bedside. Finally, the last set of triple nerve blocks was performed during the afternoon rounds, following the same protocol as before.

Several advantages of favoring longer-acting regional block techniques have been reported besides those in our study. Based on work by Farrar et al., with the utilization of a continuous femoral nerve catheter, the pain score was reduced by 60% preoperatively, as well as achieving 50% and 54% lower pain scores on postoperative days 1 and 2, respectively [20].

Evidence-based medicine (EBM) plays a pivotal role in ultrasound-guided regional anesthesia, integrating rigorous research with clinical expertise and patient preferences [16]. Ultrasound techniques, by allowing the real-time visualization of nerve structures and the spread of local anesthetics, have significantly enhanced the accuracy and safety of regional anesthesia. Studies have consistently shown that ultrasound guidance leads to higher success rates, quicker onset times, and fewer complications like nerve damage compared with traditional landmark-based methods [17]. This approach not only improves patient outcomes but also supports the customization of anesthetic choices and techniques based on individual patient needs and surgical requirements.

Similarly, in the realm of CVC insertion in critically ill patients, EBM underscores the advantages of ultrasound, including both two-dimensional imaging and Doppler modalities, over traditional landmark-based techniques [18]. These ultrasound-guided methods are crucial in accurately placing catheters in veins like the internal jugular, subclavian, or femoral veins, reducing the risks of arterial puncture and other complications. Systematic reviews and meta-analyses of randomized controlled trials have demonstrated that real-time ultrasound guidance notably increases the success rate of subclavian vein cannulation and reduces complications and the number of attempts needed. While these findings suggest a significant improvement in procedural safety and efficacy, it is important to note that the evidence, although promising, is still considered to be of low certainty.

Overall, the integration of ultrasound guidance in both regional anesthesia and CVC insertion reflects the core principles of EBM. It emphasizes the use of the best available evidence, combined with clinical expertise, to optimize patient care and outcomes in these critical procedures.

Our study confirms the results of previous studies that serial femoral nerve blocks not only decrease average patient-reported pain scores and morphine consumption in the pre- and postoperative periods but also lower the rate of opioid-related side effects [14]. This is consistent with findings that controlling pain and reducing opioid requirement is associated with reductions in PONV and delirium and adds to the body of evidence that regional techniques can decrease postoperative complications.

Moreover, also found via the chart notes, patients were able to perform postoperative breathing and rehabilitation exercises [14,15]. However, unlike previous studies, we could not find that more patients were discharged with or without home health services [14]. 

The present study aimed to assess the efficacy of single-injection nerve blocks (SINBs) for managing pain during BHA surgery. Although the study showed that SINBs provided effective pain relief, they had certain limitations that require further exploration.

One such limitation of our study was its retrospective nature, which underscores the need for a randomized, double-blind trial to validate the results more conclusively. Additionally, we did not consider the potential influence of antiemetic medication on postoperative pain and narcotic-related side effects. The financial aspect, particularly the cost of multiple injections required for SINBs, emerged as a limitation, especially for patients with low incomes. Furthermore, the duration of the procedure and the risk of local complications such as infection and hematoma were concerns. Importantly, this study being conducted in a single-center setting also presents a limitation, as it may affect the broader applicability of our findings to different clinical and demographic environments.

This study found that temporary weakness in the quadriceps muscles [13,14,15] caused by femoral nerve block did not hinder patients undergoing BHA surgery, as they were advised to stay in bed for 72 h, and partial weight-bearing was recommended [21,22,23,24] for several weeks after the surgery. Ankle dorsiflexion and plantarflexion exercises were also utilized to prevent deep vein thrombosis.

Although there was no statistically significant difference in hospitalization durations between the two groups, a trend toward shorter hospitalization was observed in the SINB group. Future studies could investigate pain control using continuous femoral nerve catheter dwelling or explore the impact of T-cane walking initialization day on early mobilization between the two groups.

## 5. Conclusions

Our study demonstrates that the administration of sequential lower limb nerve blocks under ultrasound guidance can significantly improve pain management in the early postoperative phase following Birmingham hip arthroplasty (BHA). This approach not only provides superior pain relief but also contributes to reducing narcotic consumption and mitigating associated adverse effects, such as postoperative nausea and vomiting (PONV) and delirium. However, it is important to recognize the limitations of our work, including its retrospective nature and single-center setting, which may impact the generalizability of our findings. Future research should focus on prospective, multicenter studies to validate these results and explore the long-term effects of this pain management strategy. Additionally, investigating the economic implications of this approach, especially in different healthcare systems, would provide valuable insights into its feasibility and broader applicability.

## Figures and Tables

**Figure 1 jcm-13-00338-f001:**
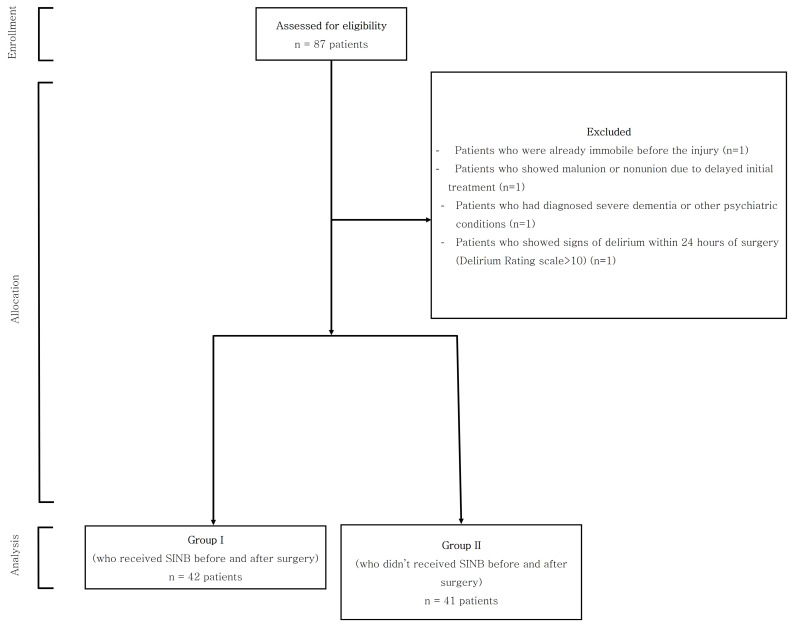
Flow chart of patients’ registration.

**Table 1 jcm-13-00338-t001:** Patient demographics.

	I ^‡^ (*n* = 42)	II ^§^ (*n* = 41)	*p*-Value
Mean age (years)	76.24	77.49	0.236
Gender			0.846
Male	17	14	
Female	25	25	
Mean BMI * (kg/m^2^)	22.08	22.05	0.484
Mean CCI ^†^	12.45	13.36	0.26
Mean VAS measured in ward on the day of admission	6.76 (1.84)	6.87 (1.75)	0.36
Koval classification (Grade)			0.262
I	27	29	
II	6	3	
III	2	0	
IV	3	6	
V	1	0	
VI	1	2	
VII	2	1	
Mean Operation time (min)	131.57	140.15	0.005 *
Mean blood loss (mL)	343	279	0.065
Mean Urine output (mL)	360.02	279.67	0.051

* BMI: body mass index, ^†^ CCI: Charlson Comorbidity Index, ^‡^ I: 3-nerve block via continuous injection group, and ^§^ II: control group (no injection). * Significant difference between Group I and Group II.

**Table 2 jcm-13-00338-t002:** VAS * for postoperative (mean and standard deviation).

	I ^‡^ (*n* = 42)	II ^§^ (*n* = 41)	*p*-Value
6 h	1.71 (0.21)	7.38 (2.30)	0.000 *
12 h	3.81 (0.30)	6.49 (1.57)	0.000 *
24 h	1.93 (0.17)	6.41 (1.20)	0.000 *
48 h	3.45 (1.28)	5.82 (1.62)	0.000 *
72 h	4.52 (0.45)	4.62 (0.51)	0.28

* VAS: visual analog scale, ^‡^ I: 3-nerve block via serial injection group, and ^§^ II: control group (no injection). * Significant difference between Group I and Group II.

**Table 3 jcm-13-00338-t003:** PCA * consumption for 72 h postoperative (mean and standard deviation).

	I ^†^ (*n* = 42)	II ^‡^ (*n* = 41)	*p*-Value
PCA * consumption (mL)	65.72 (8.78)	83.90 (18.39)	0.000 *
Morphine equivalent dose	82.8 (4.35)	95.8 (5.12)	0.000 *
Additional analgesic injection (number of injections)	0.65	0.88	0.029 *

* PCA: patient-controlled analgesia, ^†^ I: 3-nerve block via serial injection group, and ^‡^ II: control group (no injection). * Significant difference between Group I and Group II.

**Table 4 jcm-13-00338-t004:** Complications, lengths of stays, and ambulation function.

	I * (*n* = 42)	II ^†^ (*n* = 41)	*p*-Value
General complications			
Pressure sore	1	2	0.571
Pneumonia	2	3	0.875
DVT ^‡^ postoperative nausea and vomiting delirium	0	0	-
	0	6	0.013 *
	3	10	0.032 *
Local complications	0	-	-
Length of stay (day) (mean)	39.26	44.54	0.073
T-cane walking (day) (mean)	11.52	11.97	0.40

***** I: 3-nerve block via serial injection group, ^†^ II control group (no injection), and ^‡^ DVT: deep vein thrombosis. * Significant difference between Group I and Group II.

## Data Availability

The datasets generated during and/or analyzed during the current study are not publicly available due to the IRB (“Public Institutional Review Board Designated by Ministry of Health and Welfare”) guideline, where it is stipulated that the patients’ personal data should be discarded within 6 months after data collection. If data are needed, please contact us.

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
