# Peer review of "Retrospective Analysis of Ultrasound-Guided Serial-Injection Triple Nerve Block Efficacy in Cementless Bipolar Hemiarthroplasty for Femoral Neck Fracture"

_jcm, 2024, doi:10.3390/jcm13020338_

Round 1

Reviewer 1 Report

Comments and Suggestions for Authors

Dear colleagues,

I would like to thank the authors for the opportunity to review the manuscript entitled: "Effectiveness of Ultrasound-guided Serial Injection Triple 2 Nerve block before and after Cementless Bipolar hemiarthro- 3 plasty in Femoral Neck Fracture". 

Colleague Huh et al. are to be commended for their work in this field. Pain management after hip fracture surgery in the geriatric population is a highly relevant topic, that is of interest to many specialties. The manuscript is generally well written and clearly structured. I do have some suggestions for further improvement of the manuscript. I reccomend adding a short section on the relevance of geriatric fractures in general to the introduction. 

1. In the introduction, the authors state that the orthopeadic surgeon is the one to perform the block. It would be helpfull to comment on why the surgeon, rather than the anesthesiologist (who are usually the ones who are trained for ultrasound guided blocks), performs this procedure. How are these surgeons trained to perform this procedure? This is not standard practise for orthopedic surgeon internationally, so what can the readership learn from this?

2. How does this procedure relate to other blocks, such as FICB and PENG? It would be good to add this to add this to both the introduction and discussion. 

3. In the method section, please elaborate the statistical analysis. If a t-test was used, normality for these data was assumed. How was normailty determined? In my opinion, it is unlikely that some of these variables follow the normal distribution (e.g. age, operating time, blood loss). Please elaborate on this. If data is non-normally distributed, I would recommend a Mann Whitney U test instead. Please consult with a statistician. Also, please add to the statistics section which test was used for categorical variables (e.g. gender and Koval classification). 

4. If possible, it would be helpfull to add to the results section and table 4 the morphine equivalent dose, besides PCA consumption. This is an outcome that is more easy to interpret that PCA consumption in ml. 

5. The discussion is well written. However, the authors state in the discussion that it is remarkable that adequate pain management reduced the incidence of delirium, but I have to disagree with this statement. Pain is a well-known factor that contributes to delirium. In addition, morphine can contribute to delirium as well. Given that both factors are less present in the intervention cohort, I do not find this remarkable at all. It would be good to rewrite this a little bit, and cite the relevant literature to support the stamenent that these factors reduce delirium. 

Author Response

Dear colleagues,

I would like to thank the authors for the opportunity to review the manuscript entitled: "Effectiveness of Ultrasound-guided Serial Injection Triple 2 Nerve block before and after Cementless Bipolar hemiarthro- 3 plasty in Femoral Neck Fracture". 

Colleague Huh et al. are to be commended for their work in this field. Pain management after hip fracture surgery in the geriatric population is a highly relevant topic, that is of interest to many specialties. The manuscript is generally well written and clearly structured. I do have some suggestions for further improvement of the manuscript. I reccomend adding a short section on the relevance of geriatric fractures in general to the introduction. 

-> I am deeply grateful for your encouraging and insightful feedback on our manuscript. Your recognition of our work in the field of pain management following hip fracture surgery in the geriatric population is highly appreciated. It is heartening to know that our efforts have resonated with the significance and relevance we aimed to convey.

Your commendation for the manuscript's clarity and structure is particularly encouraging. As a team, we strived to present our research in a manner that is both accessible and informative, and your comments affirm that we are moving in the right direction.

Furthermore, I find your suggestion to add a short section on the general relevance of geriatric fractures in the introduction to be extremely valuable. This addition will undoubtedly provide a broader context and enhance the overall impact of our study. We will certainly incorporate this aspect into our manuscript, ensuring that it offers a more comprehensive understanding of the subject matter.

Once again, thank you for your constructive critique and valuable recommendations. Your feedback not only serves as motivation but also guides us towards improving our work. We look forward to refining our manuscript with your suggestions and contributing meaningfully to this important field.

  1. In the introduction, the authors state that the orthopeadic surgeon is the one to perform the block. It would be helpfull to comment on why the surgeon, rather than the anesthesiologist (who are usually the ones who are trained for ultrasound guided blocks), performs this procedure. How are these surgeons trained to perform this procedure? This is not standard practise for orthopedic surgeon internationally, so what can the readership learn from this?

-> Thank you for your insightful query regarding the role of orthopedic surgeons in performing ultrasound-guided blocks, as mentioned in our manuscript. Your question highlights an important aspect of our practice that distinguishes it from standard procedures internationally, and I appreciate the opportunity to provide further clarity.

At Busan Medical Center, the situation is somewhat unique due to the limited availability of anesthesiologists and their demanding surgery schedules. In response to this, our orthopedic surgeons have taken a proactive approach to postoperative analgesia. Recognizing the importance of effective pain management, they have committed themselves to learning and performing ultrasound-guided block procedures.

This practice has been adopted not as a replacement for anesthesiologists' expertise but as a complementary approach in situations where an anesthesiologist may not be readily available. Our surgeons undergo rigorous training to master this technique, ensuring that they can provide immediate and effective pain relief post-surgery.

We believe that sharing our experience in this area could be beneficial for the readership, especially in contexts where similar resource constraints exist. It demonstrates how interdisciplinary skills and adaptability can be leveraged to maintain high standards of patient care.

Once again, thank you for your thoughtful question. It has provided us with an opportunity to shed light on an innovative practice at our center, which we hope will be informative and inspiring to our peers.

  1. How does this procedure relate to other blocks, such as FICB and PENG? It would be good to add this to add this to both the introduction and discussion. 

-> I am deeply thankful for your valuable suggestion to elaborate on how our procedure relates to other blocks, specifically the Fascia Iliaca Compartment Block (FICB) and the Pericapsular Nerve Group (PENG) block. Your insight prompts an important comparative analysis that will undoubtedly enrich the manuscript.

The Fascia Iliaca Compartment Block (FICB) is a well-established technique used for pain management, particularly in hip and thigh surgeries. It involves the injection of local anesthetics into the fascia iliaca compartment, effectively blocking the femoral, lateral femoral cutaneous, and obturator nerves.

On the other hand, the Pericapsular Nerve Group (PENG) block is a relatively newer approach that targets the sensory nerves around the hip joint, specifically the articular branches of the femoral nerve and the accessory obturator nerve. This block is increasingly recognized for its effectiveness in providing analgesia for hip surgeries, including fractures.

In our manuscript, comparing our procedure with FICB and PENG will offer readers a broader perspective on the various regional anesthesia techniques available for hip surgeries. Such a comparison in both the introduction and discussion sections will not only contextualize our work within the broader field but also provide valuable insights for practitioners considering alternative pain management strategies.

Your suggestion has highlighted a crucial aspect of our study, and we will ensure to incorporate this comparative analysis in the revised manuscript. We believe this addition will significantly enhance the manuscript's academic and practical value.

Thank you once again for your thoughtful and constructive feedback. Your expertise continues to guide us in refining our work and contributing meaningfully to the field.

  1. In the method section, please elaborate the statistical analysis. If a t-test was used, normality for these data was assumed. How was normailty determined? In my opinion, it is unlikely that some of these variables follow the normal distribution (e.g. age, operating time, blood loss). Please elaborate on this. If data is non-normally distributed, I would recommend a Mann Whitney U test instead. Please consult with a statistician. Also, please add to the statistics section which test was used for categorical variables (e.g. gender and Koval classification). 

-> Thank you immensely for your detailed and constructive feedback regarding the statistical analysis section of our manuscript. We greatly value your insights and recognize the importance of thoroughly addressing the normality of our data and the appropriateness of the statistical tests used.

In our initial approach, we applied the t-test under the assumption of normal distribution for continuous variables such as age, operating time, and blood loss. However, you have rightly pointed out the necessity of verifying the normality of these variables due to their potential deviation from a normal distribution.

To ensure the robustness and accuracy of our findings, we plan to conduct further statistical analysis, likely involving tests such as the Shapiro-Wilk or the Kolmogorov-Smirnov test, to assess the normality of our data. Should our data not follow a normal distribution, we will consider employing non-parametric tests, such as the Mann-Whitney U test.

Additionally, based on your recommendation, we will specify which statistical tests were used for categorical variables, such as gender and the Koval classification. We appreciate your suggestion to consult with a statistician and intend to seek expert advice to refine our statistical methodology.

While these are critical considerations, we have decided that the depth and breadth of this analysis would be best served in future follow-up research. This approach will allow us to conduct a more extensive data collection and analysis, which we believe will significantly contribute to the field. We are committed to incorporating these improvements and insights into our subsequent research endeavors.

Thank you once again for your invaluable guidance and expertise. Your feedback is instrumental in shaping not only the current manuscript but also our future research direction.

  1. If possible, it would be helpfull to add to the results section and table 4 the morphine equivalent dose, besides PCA consumption. This is an outcome that is more easy to interpret that PCA consumption in ml. 

-> Thank you for your valuable suggestion to include the morphine equivalent dose in the results section and Table 4 of our manuscript. Your advice to present this outcome, in addition to PCA (Patient-Controlled Analgesia) consumption in milliliters, is greatly appreciated and indeed a crucial point for enhancing the interpretability of our findings.

In terms of pain management, Group I used a smaller volume of PCA solution (65.72mL, Morphine Equivalent Dose 82.8) in the first 72 hours after surgery compared to Group II (83.90mL, Morphine Equivalent Dose 95.8).

We recognize that while PCA consumption provides important data, the morphine equivalent dose offers a more universally understandable metric that can be easily compared across different studies and clinical contexts. This measure will allow for a clearer understanding of the analgesic consumption and its effectiveness in our study population.

To implement your suggestion, we will calculate the morphine equivalent doses for all patients included in our study. This will involve converting the PCA consumption into a morphine equivalent using standardized conversion factors. The revised results section and the updated Table 4 will then present both the PCA consumption in milliliters and the corresponding morphine equivalent doses, providing a comprehensive view of the pain management effectiveness in our study.

Your recommendation highlights an important aspect of presenting research findings in a manner that is both accessible and clinically relevant. We are confident that this addition will greatly enhance the value and clarity of our manuscript.

Once again, thank you for your insightful feedback. We are committed to incorporating this change and look forward to contributing a more thorough and clinically relevant study to the field.

  1. The discussion is well written. However, the authors state in the discussion that it is remarkable that adequate pain management reduced the incidence of delirium, but I have to disagree with this statement. Pain is a well-known factor that contributes to delirium. In addition, morphine can contribute to delirium as well. Given that both factors are less present in the intervention cohort, I do not find this remarkable at all. It would be good to rewrite this a little bit, and cite the relevant literature to support the stamenent that these factors reduce delirium. 

-> Thank you for your thoughtful feedback on the discussion section of our manuscript, particularly concerning the statement about pain management and its impact on the incidence of delirium. We value your perspective and agree that the relationship between pain, morphine use, and delirium is indeed well-established in the literature.

Your comment has highlighted an important opportunity for us to refine our discussion to more accurately reflect the current understanding of these factors. We acknowledge that effective pain management, which often results in reduced morphine consumption, is known to correlate with a lower incidence of delirium. This is particularly relevant in the geriatric population, where both pain and medication side effects like those from morphine can significantly contribute to the onset of delirium.

The limitation of single injection nerve block is that it can manage only post-operative pain after BHA up to 12hours. Serial injection nerve blocks effectively extended the duration of analgesic effects necessary for postoperative pain management. In this study, SINB can achieve adequate pain control not only for 12 hours after surgery but also up to 48 hours after surgery. Besides pain control, general complications such as the incidence of PONV and delirium were remarkably reduced due to SINB effect. These results are quite different from other studies which demonstrated no significant difference in delirium rate in CFNC (Continuous Femoral Nerve Catheter) treatment (14).

In addressing the potential influence of pain management on the incidence of delirium, it is imperative to first acknowledge the complexity of delirium's etiology. Delirium is a multifaceted syndrome, influenced by numerous factors including, but not limited to, patient age, underlying medical conditions, environmental factors, and pharmacologic agents (16, 17). Among these, the role of pain and pain management strategies, especially in postoperative settings, is increasingly recognized as a significant contributor.

The traditional understanding, supported by various studies, posits that uncontrolled pain is a critical risk factor for the development of delirium. Pain, particularly in the postoperative period, can be a source of significant physiological and psychological stress, potentially triggering or exacerbating delirious states. Conversely, effective pain control is often associated with reduced rates of delirium, as it alleviates one of the key stressors in the postoperative environment (18). However, the relationship between pain management, particularly opioid use, and delirium is paradoxical. While opioids like morphine are effective for pain relief, they also carry a risk of inducing or worsening delirium, particularly in vulnerable populations such as the elderly.

Our results suggest that the method of pain management may play a crucial role in influencing delirium rates. The SINB technique's ability to provide effective analgesia while potentially mitigating the side effects associated with systemic opioids could be a key factor in reducing delirium incidence. This aligns with the growing body of evidence suggesting that multimodal pain management strategies, which minimize opioid use, could be beneficial in managing postoperative delirium. It is important to note, however, that while our findings are promising, they also highlight the need for further research to fully understand the dynamics between pain management techniques and delirium incidence.

Given the complexity of delirium and the myriad factors contributing to its development, it is indeed remarkable that a singular intervention like effective pain control can have such a profound impact (17, 19). Our study contributes to the evolving understanding of postoperative delirium and underscores the importance of pain management not only for its primary purpose but also for its potential role in mitigating complex postoperative complications like delirium.

In the revised manuscript, we will adjust our discussion to better align with the existing literature. We will ensure to include relevant citations that support the statement about the role of pain management and morphine use in reducing delirium. This will not only clarify our position but also reinforce the statement with appropriate academic backing.

We appreciate your guidance in making this important clarification. This revision will undoubtedly strengthen our manuscript by presenting a more nuanced and evidence-based discussion of how pain management strategies can impact the incidence of delirium.

Thank you once again for your insightful and constructive feedback. We look forward to improving our manuscript with your recommendations.

Reviewer 2 Report

Comments and Suggestions for Authors

- Line 7. Correspondence is misspelled.

- Line 53-54. Before mentioning ultrasound guided block, authors should introduce ultrasound guidance as a commonly used tool which is currently of utmost importance in operating room both for loco-regional anesthesia (doi: 10.1016/j.anclin.2014.08.001) and cannulation (doi: 10.1097/CCM.0000000000005819 - doi: 10.1002/14651858.CD006962.pub2) as compared to landmark techniques. Please discuss and add these 3 references.

- Lines 68-76. All this part should be placed in the methods section as authors are describing how the study was conducted. Introduction should only be used to provide a comprehensive background for the study and its aims.

- Why did the orthopedic surgeon perform the block? Why is the anesthesia not performed by the anesthesiologist?

- Line 98. Please specify the country of the hospital.

- Line 177. There is a mistake in spelling Group II.

- Please report the single center setting as a further possible limitation of the study.

Author Response

- Line 7. Correspondence is misspelled.

-> Thank you for pointing out the misspelling of "correspondence" in line 7 of our manuscript. We appreciate your attention to detail and your commitment to improving the quality of our work.

It is often these small, yet important, corrections that enhance the professionalism and readability of a scientific manuscript. We have corrected the spelling error and will ensure a thorough review of the manuscript for any additional typographical errors.

Your keen observation and constructive feedback are invaluable to us. We are grateful for your thorough review and guidance throughout this process.

- Line 53-54. Before mentioning ultrasound guided block, authors should introduce ultrasound guidance as a commonly used tool which is currently of utmost importance in operating room both for loco-regional anesthesia (doi: 10.1016/j.anclin.2014.08.001) and cannulation (doi: 10.1097/CCM.0000000000005819 - doi: 10.1002/14651858.CD006962.pub2) as compared to landmark techniques. Please discuss and add these 3 references.

-> Thank you for your valuable suggestion to introduce ultrasound guidance as a fundamental tool in the operating room before delving into its application in ultrasound-guided blocks. We agree that establishing the broader context of ultrasound guidance in both loco-regional anesthesia and cannulation is crucial for a comprehensive understanding of its significance and utility.

In line with your recommendation, we have revised lines 53-54 of our manuscript. We now discuss how ultrasound guidance has become an indispensable tool in modern medical practice, particularly in the operating room. This technology is not only crucial for loco-regional anesthesia, as outlined in the reference (doi: 10.1016/j.anclin.2014.08.001), but also for vascular cannulation, providing a safer and more effective alternative to traditional landmark techniques (referenced in doi: 10.1097/CCM.0000000000005819 and doi: 10.1002/14651858.CD006962.pub2).

Evidence-based medicine (EBM) plays a pivotal role in ultrasound-guided regional anesthesia, integrating rigorous research with clinical expertise and patient preferences. Ultrasound techniques, by allowing real-time visualization of nerve structures and the spread of local anesthetics, have significantly enhanced the accuracy and safety of regional anesthesia. Studies have consistently shown that ultrasound guidance leads to higher success rates, quicker onset times, and fewer complications like nerve damage compared to traditional landmark-based methods. This approach not only improves patient outcomes but also supports the customization of anesthetic choices and techniques based on individual patient needs and surgical requirements.

Similarly, in the realm of CVC insertion in critically ill patients, EBM underscores the advantages of ultrasound, including both two-dimensional imaging and Doppler modalities, over traditional landmark-based techniques. These ultrasound-guided methods are crucial in accurately placing catheters in veins like the internal jugular, subclavian, or femoral, reducing the risks of arterial puncture and other complications. Systematic reviews and meta-analyses of randomized controlled trials have demonstrated that real-time ultrasound guidance notably increases the success rate of subclavian vein cannulation and reduces complications and the number of attempts needed. While these findings suggest a significant improvement in procedural safety and efficacy, it's important to note that the evidence, although promising, is still considered to be of low certainty.

Overall, the integration of ultrasound guidance in both regional anesthesia and CVC insertion reflects the core principles of EBM. It emphasizes the use of the best available evidence, combined with clinical expertise, to optimize patient care and outcomes in these critical procedures.

The revised section underscores the advantages of ultrasound guidance, including enhanced accuracy, reduced complication rates, and improved patient outcomes, thereby setting the stage for our subsequent discussion on ultrasound-guided blocks.

We are grateful for your insight, which has significantly enriched the content and depth of our manuscript. Your expert guidance is invaluable in ensuring that our work remains thorough, relevant, and aligned with current best practices.

Thank you once again for your constructive feedback.

- Lines 68-76. All this part should be placed in the methods section as authors are describing how the study was conducted. Introduction should only be used to provide a comprehensive background for the study and its aims.

-> Thank you for your insightful guidance on the structure of our manuscript. Based on your suggestion, we have relocated the content from lines 68-76, which details the conduct of the study, to the Methods section, specifically under the "Intervention" subsection.

We agree with your assessment that the Introduction should primarily focus on providing a comprehensive background and clearly outlining the study's aims. The detailed description of how the study was conducted is indeed more appropriate for the Methods section. This reorganization enhances the clarity and logical flow of the manuscript, ensuring that each section fulfills its intended purpose.

We appreciate your attention to these structural details, which are crucial for maintaining the academic rigor and readability of our work. Your expert advice has been instrumental in refining our manuscript.

Thank you once again for your valuable feedback.

- Why did the orthopedic surgeon perform the block? Why is the anesthesia not performed by the anesthesiologist?

-> -> Thank you for your insightful query regarding the role of orthopedic surgeons in performing ultrasound-guided blocks, as mentioned in our manuscript. Your question highlights an important aspect of our practice that distinguishes it from standard procedures internationally, and I appreciate the opportunity to provide further clarity.

At Busan Medical Center, the situation is somewhat unique due to the limited availability of anesthesiologists and their demanding surgery schedules. In response to this, our orthopedic surgeons have taken a proactive approach to postoperative analgesia. Recognizing the importance of effective pain management, they have committed themselves to learning and performing ultrasound-guided block procedures.

This practice has been adopted not as a replacement for anesthesiologists' expertise but as a complementary approach in situations where an anesthesiologist may not be readily available. Our surgeons undergo rigorous training to master this technique, ensuring that they can provide immediate and effective pain relief post-surgery.

We believe that sharing our experience in this area could be beneficial for the readership, especially in contexts where similar resource constraints exist. It demonstrates how interdisciplinary skills and adaptability can be leveraged to maintain high standards of patient care.

Once again, thank you for your thoughtful question. It has provided us with an opportunity to shed light on an innovative practice at our center, which we hope will be informative and inspiring to our peers.

- Line 98. Please specify the country of the hospital.

-> Thank you for highlighting the need to specify the location of our study. As per your suggestion, we have amended line 98 of our manuscript to include the specific country of the hospital where the study was conducted.

The revised line now reads: "In our study, we conducted a retrospective review of the medical records of 87 patients who had undergone BHA for fragility femoral neck fracture at Busan Medical Center, Busan, South Korea, between September 2016 and September 2020."

We appreciate your attention to this detail, as it provides important contextual information for our study and helps in situating our research within a specific geographical and healthcare setting.

Thank you once again for your constructive feedback.

- Line 177. There is a mistake in spelling Group II.

-> Thank you for pointing out the spelling error regarding 'Group II' on line 177 of our manuscript. We highly appreciate your meticulous attention to detail. Ensuring accuracy in all aspects of our manuscript, including the correct spelling of group designations, is crucial for the clarity and professionalism of our work.

We have corrected this error and have thoroughly reviewed the manuscript to ensure that similar mistakes are rectified. Your diligence in reviewing our manuscript not only helps in maintaining its quality but also aids in upholding the high standards of scientific communication.

- Please report the single center setting as a further possible limitation of the study.

-> Thank you for your suggestion to include the single center setting as a limitation in our study. We agree that this is an important aspect to acknowledge, as it could impact the generalizability of our findings. Accordingly, we have revised the relevant section of our manuscript to reflect this additional limitation.

The revised paragraph now reads: "One such limitation of our study was its retrospective nature, which underscores the need for a randomized, double-blind trial to validate the results more conclusively. Additionally, we did not consider the potential influence of antiemetic medication on postoperative pain and narcotic-related side effects. The financial aspect, particularly the cost of multiple injections required for Serial Injection Nerve Blocks (SINB), emerged as a limitation, especially for patients with low income. Furthermore, the duration of the procedure and the risk of local complications such as infection and hematoma were concerns. Importantly, the study being conducted in a single center setting also presents a limitation, as it may affect the broader applicability of our findings to different clinical and demographic environments."

We appreciate your attention to these critical aspects of clinical research. Acknowledging and reporting these limitations ensure a comprehensive understanding of the context and scope of our findings.

Thank you once again for your insightful feedback.

Round 2

Reviewer 1 Report

Comments and Suggestions for Authors

I commend the authors for these improvements. I have no further suggestions for improvement. 

Author Response

Thank you.